# SOUNDSTORM: EFFICIENT PARALLEL AUDIO GENERATION

## ABSTRACT

Modeling the tokens of a neural audio codec unlocked rapid progress in audio generation, producing high-quality, coherent audio. However, this approach requires modeling long sequences, thus affecting the training and inference costs. In this work, we propose SoundStorm, a model for efficient, parallel audio generation, which scales gracefully to long sequences without compromising the quality of the generated audio. SoundStorm receives as input coarse, discrete audio representations, and relies on bidirectional attention and confidence-based parallel decoding to sample the tokens of a neural audio codec. Compared to the autoregressive generation approach of AudioLM, our model produces audio of the same quality and with higher consistency in voice and acoustic conditions, while being two orders of magnitude faster. SoundStorm generates 30 seconds of audio in 0.5 seconds on a TPU-v4. We also demonstrate the ability of our model to synthesize high-quality, natural dialogue segments, given a transcript annotated with speaker turns and a short prompt with the speakers' voices. Audio samples are available in the supplementary material.

## 1 INTRODUCTION

Modeling discrete representations of audio produced by neural codecs (Zeghidour et al., 2022; Défossez et al., 2022) makes the task of audio generation amenable to the powerful Transformer-based sequence-to-sequence modeling approaches (Vaswani et al., 2017). Casting unconditional and conditional audio generation as sequence-to-sequence modeling has unlocked rapid progress in speech continuation (Borsos et al., 2023), text-to-speech (Wang et al., 2023; Kharitonov et al., 2023), and general audio and music generation (Kreuk et al., 2023; Agostinelli et al., 2023; Copet et al., 2023).

For generating high-quality audio by modeling the tokens of a neural codec, the rate of the discrete representation must be increased, resulting in either an exponential growth in the codebook size or in long token sequences. While the exponential growth of the codebook is prohibitive due to memory limitations, in turn, long token sequences also present computational challenges for autoregressive models. In particular, Transformer-based models will incur quadratic runtime complexity with respect to the sequence length for calculating the self-attention. Thus, addressing the trade-off between perceptual quality and runtime is one of the core challenges for audio generation.

The problem of generating long audio token sequences can be addressed by at least three orthogonal approaches, or a combination thereof: i) efficient attention mechanisms (Kitaev et al., 2020; Choromanski et al., 2021; Xiong et al., 2021; Hawthorne et al., 2022), ii) non-autoregressive, parallel decoding schemes (Gu et al., 2018; Ghazvininejad et al., 2019; Chang et al., 2022), iii) custom architectures adapted to the special structure of the tokens produced by neural audio codecs (Kreuk et al., 2023; Wang et al., 2023; Lee et al., 2022; Agostinelli et al., 2023). However, in the context of modeling the token sequence of neural audio codecs, either unconditionally or based on weak conditioning such as text, the naive flattened modeling of the tokens typically outperforms custom modeling schemes, leaving the efficient generation of long, high-quality audio segments an open problem.

The special structure of the audio token sequence holds promise for future advances in long-sequence audio modeling. Concretely, both SoundStream (Zeghidour et al., 2022) and EnCodec (Défossez et al., 2022) rely on Residual Vector Quantization (RVQ), where each compressed audio frame is quantized by a series of quantizers, with each quantizer operating on the residual of the previous one, and the number of quantizers controlling the overall bitrate. This induces a hierarchical token

structure, where tokens from finer RVQ levels contribute less to the perceptual quality, allowing for efficient factorizations and approximations of the joint distribution of the token sequence. Hence, the models and decoding schemes should take this special structure of the input into account for efficient training and inference.

In this work, we present SoundStorm, a model for efficient and high-quality audio generation. SoundStorm addresses the problem of generating long audio token sequences by relying on: i) an architecture adapted to the hierarchical structure of the audio tokens, ii) a parallel, non-autoregressive, confidence-based decoding scheme inspired by MaskGIT (Chang et al., 2022) for residual vector-quantized token sequences.

SoundStorm relies on a bidirectional attention-based Conformer (Gulati et al., 2020) that is trained to predict masked audio tokens produced by a residual vector-quantized neural audio codec, given a coarse, low-bitrate conditioning signal such as the semantic tokens of AudioLM (Borsos et al., 2023). On the input side, it aggregates the embeddings of the audio tokens corresponding to the same time frame, such that the internal sequence length for the self-attention is identical to the number of times frames, and independent of the number of quantizers in the RVQ. The output embeddings are then processed by separate heads per RVQ level to predict the masked target tokens. At inference time, given the conditioning signal, SoundStorm starts with all audio tokens masked out, and fills in the masked tokens RVQ level-by-level over several iterations, predicting multiple tokens in parallel during a single iteration within a level. To support this inference scheme, we propose a masking scheme for training that mimics the inference procedure.

We demonstrate that SoundStorm can serve as AudioLM's acoustic generator, replacing both AudioLM's stage two (coarse acoustic model) and stage three (fine acoustic model). SoundStorm produces audio two orders of magnitude faster than AudioLM's hierarchical autoregressive acoustic generator with matching quality and improved consistency in terms of speaker identity and acoustic conditions. Furthermore, we show that SoundStorm, coupled with the text-to-semantic modeling stage of SPEAR-TTS (Kharitonov et al., 2023), can synthesize high-quality, natural dialogues, allowing one to control the spoken content (via transcripts), speaker voices (via short voice prompts) and speaker turns (via transcript annotations). When synthesizing dialogues of 30 seconds, we measure a runtime of 2 seconds on a single TPU-v4 (Jouppi et al., 2023).

## 2 RELATED WORK

**Modeling the representations of neural audio codecs.** Unsupervised speech embeddings (Baevski et al., 2020; Hsu et al., 2021; Chung et al., 2021) have provided a low-framerate representation of the underlying signal which remains rich enough after discretization for language models to generate intelligible speech from a specific speaker as a sequence of tokens (Lakhotia et al., 2021). Neural audio codecs (Zeghidour et al., 2022; Défossez et al., 2022), with their ability of reconstructing high-quality audio at very low bitrates, subsequently allowed for performing generation in their continuous latent spaces (Shen et al., 2023), as well as in their discrete token spaces, in domains as diverse as multi-speaker speech and piano (Borsos et al., 2023; Kharitonov et al., 2023), music (Agostinelli et al., 2023) or sound effects (Kreuk et al., 2023).

In particular, AudioLM (Borsos et al., 2023) introduces a hierarchical sequence-to-sequence approach where high-level semantic tokens are generated as an intermediate representation, which is then used as a conditioning signal for predicting tokens of a SoundStream (Zeghidour et al., 2022) codec. While this hierarchical approach has demonstrated remarkable results for speech (Kharitonov et al., 2023) and music modeling (Agostinelli et al., 2023; Donahue et al., 2023), the computational cost of modeling flattened SoundStream tokens with self-attention scales quadratically with the sequence length and thus the bitrate of the neural codec, preventing these models from generating long-form audio with high quality. SoundStorm alleviates this issue by modeling the multi-level tokens of the neural codec in parallel, inducing a two-order of magnitude speed-up over autoregressive modeling and unlocking the ability to scale audio generation abilities both in quality and in sequence length.

**RVQ-aware architectures.** A common design choice for modeling RVQ token sequences is to sum the embeddings corresponding to the same RVQ input embedding (frame) in order to reduce the sequence length. Operating on such sequences, AudioGen (Kreuk et al., 2023) proposes a Transformer with $Q$ separate heads for the different RVQ levels, predicting the tokens for an RVQ

frame in parallel. While providing a significant speedup for inference, the authors found that, for text-to-audio generation, this approach has an inferior performance compared to modeling the token sequence of a neural audio codec with similar bitrate and reconstruction quality, but with a single level of quantization. MusicGen (Copet et al., 2023) proposes predicting a delayed RVQ pattern in parallel, which benefits from efficient inference but does not match the quality of flattened modeling.

VALL-E (Wang et al., 2023) instead relies on a hybrid approach, where the tokens corresponding to the first RVQ level are predicted autoregressively, and the subsequent levels are produced non-autoregressively. The latter is achieved by a model that sums up the embeddings from the same RVQ input frame, and applies bidirectional self-attention to predict all tokens from RVQ level $q + 1$ given all tokens from levels $1, \dots, q$, the acoustic prompt and the phoneme sequence. During inference, tokens starting from the second level of the RVQ are produced iteratively, performing greedy decoding (choosing the most likely tokens) level-by-level. Level-wise greedy decoding represents the baseline for our method.

Modeling sequences produced by RVQ has been also investigated in domains other than audio. For example, the RQ-Transformer (Lee et al., 2022) also adds up the embeddings corresponding to the same RVQ input frame, but factorizes the full joint distribution efficiently with a spatial and a depth Transformer, for modeling autoregressively the RVQ frames and tokens within the frames, respectively.

**Parallel decoding.** In order to improve the inference time and to allow bidirectional non-causal attention on the input sequence, parallel decoding schemes have been proposed for text (Gu et al., 2018; Ghazvininejad et al., 2019), image (Chang et al., 2022) and video generation (Villegas et al., 2023). Of particular relevance to our work is the parallel, iterative sampling scheme of MaskGIT (Chang et al., 2022). During inference time, MaskGIT starts from masked tokens, and in each round, predicts a portion of the tokens based on confidence scores. The portion of the predicted tokens in each round is controlled by a schedule, and usually progressively increases over the iterations — once predicted, the tokens are treated as fixed. Our proposed decoding scheme can be seen as the extension of MaskGIT's decoding to token sequences produced by residual quantization. We note that, in a parallel work to ours, Garcia et al. (2023) also propose a MaskGIT-based parallel decoding scheme for music generation.

## 3 METHOD

SoundStorm receives as input a sequence of discrete tokens representing the conditioning signal and produces as output a sequence of SoundStream tokens, which can be decoded back to audio waveforms. We assume that the conditioning signal is time-aligned with the SoundStream frames or can be upsampled to the same rate. Such a conditioning signal is, for example, the semantic token sequence used in AudioLM, SPEAR-TTS or MusicLM, which makes our method a drop-in replacement for the acoustic generators of these models. We leave the extension to other types of conditioning signals via cross-attention or to unconditional sampling for future work, and focus our presentation of SoundStorm as the acoustic generator within AudioLM, replacing both AudioLM's coarse and fine acoustic modeling stages.

### 3.1 ARCHITECTURE

The architecture of the model is illustrated in Figure 1. At the input side, we interleave the time-aligned conditioning tokens with the SoundStream tokens at the frame level, embed the resulting sequence, sum the embeddings corresponding to the same frame, including the embedding of the conditioning token, and pass the resulting continuous embeddings to a Conformer. Consequently, the sequence length for bidirectional self-attention in the Conformer is determined by the number of SoundStream frames (typically 50 per second), and thus is independent of the number of RVQ levels $Q$, allowing one to handle audio with length on the order of minutes. At the output side, we use $Q$ dense layers as heads to produce the target SoundStream tokens.

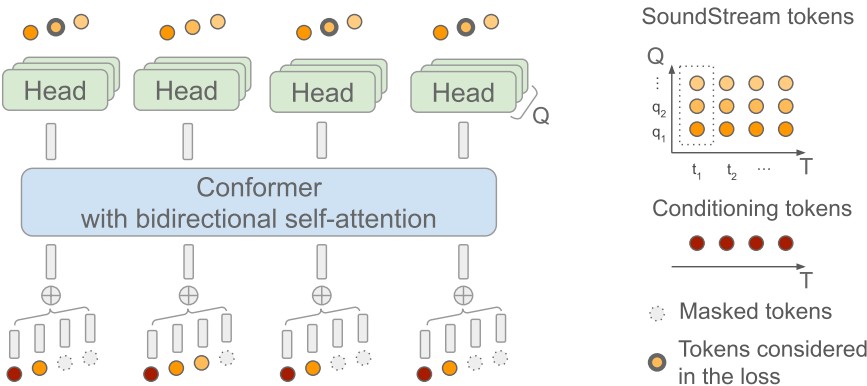

Figure 1: SoundStorm architecture and masking scheme for training (without prompting). The model reduces the input sequence length by summing up the embeddings of the tokens corresponding to the same SoundStream frame. During training, an RVQ level $q$ is sampled ($q = 2$ out of $Q = 3$ levels in the figure), and a subset of randomly sampled tokens at level $q$ are masked together with all tokens at RVQ levels $q + 1, \ldots, Q$. The loss is computed only on the masked tokens at level $q$.

## 3.2 MASKING

Before presenting our masking strategy, let us introduce some notation and definitions. Let $S_t$ denote the SoundStream embedding at timestep $t$, $Y \in \{1, \ldots, C\}^{T \times Q}$ denote the SoundStream tokens, where $C$ indicates the codebook size used in each RVQ level out of the $Q$ levels, and $E_{q,c}$ denote the embedding of SoundStream token with index $c$ at level $q$. For residual vector quantization, $S_t = \sum_{q=1}^{Q} E_{q,Y_{t,q}}$ and $Y_{t,q'} = \arg\min_i \|S_t - \sum_{q=1}^{q'-1} E_{q,Y_{t,q}} - E_{q',i}\|_2^2$, with $\|S_t - \sum_{q=1}^{Q} E_{q,Y_{t,q}}\|_2^2$ minimized during training. This introduces the coarse-to-fine hierarchical token structure within the RVQ, with coarse tokens being more important for the perceptual reconstruction quality.

For designing our masking and decoding, we extend the masking and confidence-based parallel decoding scheme of MaskGIT (Chang et al., 2022) to token sequences produced by RVQ. At a high level, our approach can be seen as following the strategy of Chang et al. (2022) per RVQ level in a coarse-to-fine order. The coarse-to-fine ordering is of particular importance, since it not only respects the conditional dependencies between levels of the RVQ hierarchy, but also exploits the conditional independence of tokens from finer levels given all tokens from coarser levels. The tokens of finer levels are responsible for local, fine acoustic details and can thus be sampled in parallel without a loss of audio quality.

We design our masking scheme for training accordingly. To enable voice prompting, we randomly sample a timestep $t \in \{1, \ldots, T\}$, where $T$ denotes the maximum sequence length, and we do not mask any tokens before this timestep. The conditioning tokens are never masked. Our masking scheme proceeds as follows:

- Sample the prompt delimiter timestep $t \sim \mathcal{U}\{0, T-1\}$;
- Sample the current RVQ level $q \sim \mathcal{U}\{1, Q\}$;
- Sample the mask $M \in \{0, 1\}^T$ according to a cosine schedule (Chang et al., 2022) for level $q$, i.e., sample the masking ratio $p = \cos(u)$ where $u \sim \mathcal{U}[0, \pi/2]$, and sample iid $M_i \sim \text{Bernoulli}(p)$.
- Mask the selected non-prompt tokens at the current RVQ level $q$ (mask $Y_{t',q}$ if $M_{t'} = 1$ and $t' > t$) and all non-prompt tokens at finer RVQ levels ($Y_{>t,>q}$).

Given a masked token sequence, we train the model with cross-entropy loss with the ground-truth tokens as target, where the loss is only calculated on the masked tokens within the $q$-th RVQ level. The masked token shares the same embedding in different RVQ levels. An example of this masking scheme is illustrated in Figure 1, with $T = 4$, $Q = 3$, $t = 0$ and $q = 2$. We leave the exploration

of non-uniform sampling of the RVQ level $q$ and of non-uniform loss weighting schemes based on levels for future work.

### 3.3 ITERATIVE PARALLEL DECODING

Given a conditioning signal, our decoding scheme starts with all SoundStream tokens masked out except for the ones of the prompt (if provided). Then, it proceeds to sampling the tokens RVQ level-wise in a coarse-to-fine order, only proceeding to level $q + 1$ when all tokens for levels $1, \ldots, q$ have been sampled. Within an RVQ level, we use the confidence-based sampling scheme of Chang et al. (2022). Namely, we perform multiple forward passes, and at each iteration $i$, we sample candidates for the masked positions, retaining $\text{top}_k = p_i$ of them based on confidence scores, where $p_i$ follows a cosine schedule. Compared to Chang et al. (2022), we use greedy decoding instead of confidence-based sampling for the last iteration within each RVQ level, which we found to improve the perceived audio quality.

Following the official implementation of MaskGIT, the confidence scores are determined as follows. We first sample candidates from the categorical distribution based on the logits produced by the model, with temperature of 1, for all masked positions. We retrieve the logits of the sampled candidates, to which we add Gumbel noise with a multiplier of 1 annealed linearly to 0 over the iterations based on the schedule. Then we retain the $\text{top}_k = p_i$ candidates based on the highest perturbed scores.

Performing the decoding RVQ level-wise makes it possible to exploit the conditional independence assumption along the time axis in finer levels, namely that multiple finer tokens can be sampled in parallel since they represent local, fine acoustic details. This implies that we can decrease the number of forward passes significantly as we progress to finer RVQ levels during decoding.

## 4 EXPERIMENTS

### 4.1 MODEL TRAINING AND INFERENCE SETUP

In our experiments, we rely on a SoundStream codec that produces 50 frames per second and uses an RVQ with $Q = 12$ levels, with 1024 codebook size per level, resulting in a bitrate of $50 \cdot 12 \cdot \log_2 1024 = 6000$ bps. We use the semantic tokens of AudioLM as conditioning, which originate from w2v-BERT (Chung et al., 2021) embeddings quantized with $k$-means with 1024 cluster centers. These tokens have a rate of 25 tokens per second, so we duplicate them to match the framerate of SoundStream.

We use a Conformer with 350M parameters, with 12 layers, 16 attention heads, embedding size and model dimension of 1024, feedforward dimension of 4096, convolution kernel size of 5, and rotary positional embeddings (Su et al., 2021). During decoding, we use $(16, 1, 1, \ldots, 1)$ iterations for the RVQ levels, that is, 16 iterations in the first level and greedily choosing the tokens with the highest probability in the subsequent levels, level-by-level. This strategy results in 27 forward passes with the model to predict 30 seconds of audio, or the equivalent of $30 \cdot 50 \cdot 12 = 18000$ SoundStream tokens.

We train the model on LibriLight (Kahn et al., 2020) (60k hours), with 10 epochs over the data, sampling random windows of length between 0 and 30 seconds from each example. We use a batch size of 256 (without packing), Adam with weight decay of $10^{-3}$ and learning rate of $5 \cdot 10^{-4}$, with linear rampup over 20000 steps and exponential decay over 0.5M steps.

### 4.2 SPEECH INTELLIGIBILITY, AUDIO QUALITY, VOICE PRESERVATION AND ACOUSTIC CONSISTENCY

In a series of subjective evaluation experiments, Borsos et al. (2023) and Kharitonov et al. (2023) have shown that the acoustic generation stage of AudioLM produces audio with quality indistinguishable from the quality of the ground-truth samples. Hence, we consider AudioLM's hierarchical acoustic generation stages (coarse and fine stages) as a baseline in our experiment. One important difference compared to the AudioLM acoustic stage experiments of Borsos et al. (2023) is that we require the conditioning signal to be time-aligned with the SoundStream frames, whereas the experiments of Borsos et al. (2023) use semantic tokens with duplicates removed. For a fair comparison, so that both methods use the same conditioning, we repeat the acoustic stage experiments of Borsos et al.

Table 1: Comparing intelligibility, quality, voice preservation, and acoustic consistency of AudioLM's acoustic generator and SoundStorm. We report metric values for the 'short' (4-10 s), 'mid' (10-20 s), and 'long' (20-30 s) splits of LibriSpeech test-clean separately. SoundStorm matches AudioLM's acoustic generator in terms of audio quality, and outperforms it in terms of speech intelligibility and acoustic consistency.

| | WER↓ | | | CER↓ | | | Audio quality↑ | | | Voice preservation↑ | | | Acoustic consistency↑ | | |
|---|---|---|---|---|---|---|---|---|---|---|---|---|---|---|---|
| | short | mid | long | short | mid | long | short | mid | long | short | mid | long | short | mid | long |
| Original SoundStream rec. | 2.62 | 1.95 | 2.20 | 0.89 | 0.55 | 0.69 | 3.72 | 3.91 | 3.99 | 0.63 | 0.65 | 0.66 | 0.97 | 0.95 | 0.93 |
| | *Without a speaker prompt* | | | | | | | | | | | | | | |
| AudioLM | 4.65 | 3.59 | 4.79 | 2.15 | 1.57 | 2.30 | 3.93 | 4.04 | 4.08 | – | – | – | – | – | – |
| SoundStorm | **3.48** | **2.55** | **3.33** | **1.39** | **0.89** | **1.29** | 4.01 | 4.16 | 4.20 | – | – | – | – | – | – |
| | *With a speaker prompt* | | | | | | | | | | | | | | |
| AudioLM | 3.77 | 3.40 | 3.75 | 1.50 | 1.47 | 1.54 | **3.91** | **4.06** | 4.10 | 0.46 | 0.48 | 0.48 | **0.96** | 0.91 | 0.86 |
| SoundStorm | **2.99** | **2.43** | **3.36** | **1.10** | **0.81** | **1.24** | 3.81 | 4.05 | 4.15 | 0.57 | 0.59 | 0.59 | **0.96** | **0.94** | **0.91** |

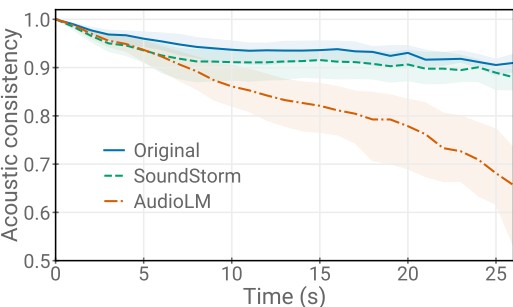

Figure 2: Acoustic consistency between the prompt and the generated audio for the samples in the 'long' split of LibriSpeech test-clean. The shaded area represents the inter-quartile range.

(2023) without removing duplicate semantic tokens, thus making the conditioning stronger. We also use the same SoundStream codec for both methods.

**Speech intelligibility.** We quantify speech intelligibility by measuring the word error rate (WER) and character error rate (CER) of the generated audio after transcribing it with ASR. The generation is conditioned on the ground-truth semantic tokens from LibriSpeech test-clean split (Panayotov et al., 2015). We perform these experiments both in the unprompted setup, where the methods can randomly sample speakers, and in the prompted setup, where the methods should respect the speaker identity provided in the form of ground-truth SoundStream tokens corresponding to the first 3-seconds. We use a Conformer Transducer-L (Gulati et al., 2020) ASR model for transcription.

We report the results separately on short (4-10 seconds), medium (10-20 seconds) and long (20-30 seconds) segments. Since AudioLM's acoustic generation stage is autoregressive in the flattened token sequence of the first 4 RVQ levels, it is prohibitively expensive to generate 30 seconds in a single pass. Hence, we generate segments longer than 10 seconds with a slide-and-prompt approach, where we generate 10-second chunks and use the last 3 seconds as the prompt for the next chunk. The results in Table 1 show that SoundStorm improves over AudioLM's acoustic generation significantly in terms of WER and CER on all splits, both for the prompted and unprompted scenario.

**Voice preservation.** Next, we measure the capability of SoundStorm to maintain the speaker identity of the prompt. To do so, we extract speaker embeddings from the prompt and the generated audio and compute their cosine similarity. As in Wang et al. (2023) and Kharitonov et al. (2023), we compute embeddings with a publicly available[1] speaker verification system based on WavLM (Chen et al., 2022). Table 1 shows that SoundStorm significantly outperforms the AudioLM baseline.

---

[1] `https://github.com/microsoft/UniSpeech/tree/main/downstreams/speaker_verification#pre-trained-models`

Table 2: Mean Opinion Score (MOS) evaluations of different models for synthesing speech based on ground truth semantic tokens, with 95% confidence intervals obtained using bootstrap.

| Model | MOS |
|---|---|
| Ground truth (SoundStream rec.) | $3.830 \pm 0.120$ |
| AudioLM flattened (stage 2&3) | $3.925 \pm 0.115$ |
| SoundStorm | $3.790 \pm 0.114$ |
| SoundStorm more iterations | $3.885 \pm 0.115$ |
| SoundStorm greedy | $2.788 \pm 0.133$ |

**Acoustic consistency drift.** In the case of long generation, it is interesting to measure to what extent the acoustic properties of the prompt (e.g., speaker identity, recording conditions) are preserved along time in the generated audio. To this end, we train a model capable of assessing whether two short segments come from the same recording. Specifically, we consider a speech corpus in which each example contains a recording from a single speaker (e.g., LibriLight) and extract two random, non-overlapping crops with duration of 2-5 seconds. Then, from each crop, we extract a sequence of embeddings from an intermediate layer of a BEST-RQ model (Chiu et al., 2022) pre-trained on the same corpus. We use layer 4 in our experiments. Each sequence of embeddings is fed to a model that consists of a Conformer layer, followed by global average pooling along the temporal dimension and a linear projection layer, so that each crop is represented with a single 1024-dimensional embedding. We train this model using the same contrastive loss as Radford et al. (2021).

To measure acoustic consistency drift, we compute the cosine similarity between the embedding computed from the prompt of 3 seconds, and the embeddings computed from subsequent crops of 3 seconds along the temporal axis. As an illustration, Figure 2 shows the measured drift on the 'long' split of LibriSpeech test-clean. We observe that for the audio generated by SoundStorm, the acoustic consistency score is close to the one measured for the original samples, while for AudioLM we observe a more significant drift over time. Table 1 reports the average acoustic consistency scores on the 'short', 'mid' and 'long' splits, where the averaging is done over non-overlapping crops. Unsurprisingly, the improvement is more evident for longer audio samples.

**Audio Quality.** For fast experimental turn-around, we use a no-reference MOS estimator similar to DNSMOS (Reddy et al., 2021) to estimate the perceived audio quality of the generated samples. The results in Table 1 show that, according to the MOS estimator, SoundStorm is on par with AudioLM's acoustic generator, which in turn has been shown to match the quality of the ground-truth audio in the subjective studies of Borsos et al. (2023) and Kharitonov et al. (2023).

To confirm these findings, we conducted a subjective listening test with 10 human raters: we selected an utterance at random of length between 8 and 10 seconds for each of the 40 speakers from `test-clean` split of LibriSpeech (Panayotov et al., 2015). Then, for each sample, we extracted the ground truth semantic tokens, and we resynthesized the sample with the various approaches in the voice-prompted setup, i.e., we sampled the rest of the acoustic tokens given all the semantic tokens and the acoustic tokens corresponding to the first 3 seconds. The raters were instructed to judge the quality of the sample on a scale of 1 to 5, taking into consideration that the first 3 seconds of each sample is non-synthetic audio — we attach the task description and the rater interface in the Appendix. We compare the SoundStream reconstruction of the ground truth audio, AudioLM's flattened acoustic generator ("AudioLM stage 2&3"), SoundStorm with the default number of iterations $(16, 1, \ldots, 1)$, level-wise greedy $(1, 1, \ldots, 1)$ and more iterations $(32, 32, 16, 1 \ldots, 1)$ — see the ablation for the number of decoding iterations in the next section. The results in Table 2 show that, except for level-wise greedy, all models match the quality of the ground truth audio.

## 4.3 RUNTIME AND ABLATIONS

**Runtime.** We measure the runtime of the different methods to generate up to 30 seconds of audio on a single TPU-v4. Figure 3 shows that SoundStorm can generate audio two orders of magnitude faster than AudioLM's acoustic generator, with a real time factor of 0.017, including decoding to waveform by the SoundStream decoder. We also measure the runtime of the semantic generation stage of AudioLM ("AudioLM Stage 1" in the figure), and conclude that by coupling the semantic

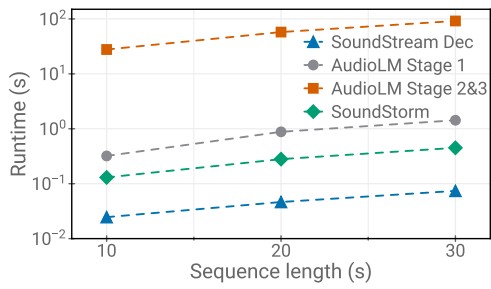 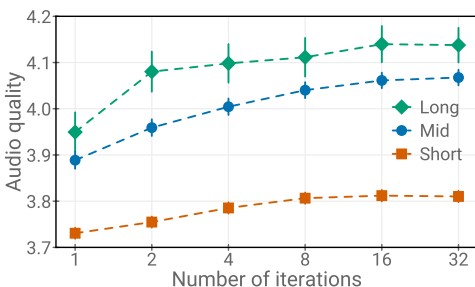

Figure 3: Runtimes of SoundStream decoding, SoundStorm and different stages of AudioLM on a TPU-v4.

Figure 4: Audio quality with respect to the number of decoding iterations in the first RVQ level.

generation stage with SoundStream, we can generate 30 seconds of speech continuation within 2 seconds (1.4 seconds for semantic generation, 0.5 seconds for SoundStorm and 0.1 seconds for SoundStream decoding).

**Number of decoding steps.** In the former experiments, we used 16 iterations for decoding the first RVQ level, and performed greedy decoding for the subsequent levels. We now investigate the effect of the number of decoding iterations for the different RVQ levels.

To achieve this, we repeat the speaker-prompted experiment on LibriSpeech test-clean with different number of decoding iterations in the first level. Figure 4 shows that, according to the audio quality estimator, our strategy of using 16 iterations achieves an increase of 0.1-0.2 in the quality score compared to level-wise greedy decoding, whereas further increasing the number of iterations does not improve the score. The artifacts produced by the greedy strategy are clearly perceivable; we provide samples produced by this strategy on the accompanying webpage. The quality impact of the greedy strategy is also confirmed by the subjective listening study in Table 2.

We have also experimented with increasing the number of iterations for RVQ levels 2-12, and we have found no statistically significant improvement in the audio quality score when synthesizing samples from LibriSpeech test-clean; this observation is confirmed both by the MOS proxy and the subjective listening study in Table 2. We note that this observation is in line with the decoding strategy of Wang et al. (2023), which is autoregressive in the first RVQ level, and level-wise greedy beyond. We hypothesize that performing multiple iterations on finer levels becomes relevant when generating audio beyond speech, where the semantic tokens and first RVQ level SoundStream tokens will not capture all the important acoustic details.

## 5    DIALOGUE SYNTHESIS

Spoken dialogue synthesis is an important application where maintaining speaker identities over multiple speaker turns and long time spans is of paramount importance. This makes it an ideal use-case for SoundStorm. In this section, we show that, by coupling SoundStorm with a text-to-semantic token model, we can synthesize high-quality multi-turn dialogues with duration of up to 30 seconds. This approach is similar to the one of SPEAR-TTS (Kharitonov et al., 2023).

We collected a corpus of approximately 100,000 hours of dialogues, segmented into 30-second chunks. By running an off-the-shelf ASR system, we generated a transcript for each segment. In these transcripts, speaker turns are annotated with a turn marker symbol in the appropriate location in the transcript. To extract semantic tokens, we train a 0.6B parameter BEST-RQ (Chiu et al., 2022) on this dataset and fit $k$-means with 4096 cluster centers to the activations of layer 13 of the BEST-RQ model. This results in 25 semantic tokens per seconds, with a codebook size of 4096. We train a SoundStream codec operating at 24 kHz, producing 50 frames per second with 12 RVQ on this corpus.

To model the text-to-semantic token mapping, we train a ByT5-large Transformer (Xue et al., 2022). This is an encoder-decoder model with 36 encoder and 12 decoder layers, embedding size of 1536 and feed-forward dimension of 3840. In total, the model has 1.2B parameters. We only train the

decoder and use a text-pretrained encoder from a published ByT5 checkpoint (Xue et al., 2022). As input, this model takes a byte-level representation of the text and predicts non-deduplicated semantic tokens. Decoding is done by temperature sampling, with temperature of 0.9 and $\text{top}_k$ set to 125. Due to the modest size of the decoder and the short target sequence length (only 750 semantic tokens for 30 seconds), running inference with this model takes 1.4 seconds on a TPU-v4.

We train both the text-to-semantic model and SoundStorm on the dialogue corpus for 10 epochs with the batch size, learning schedule and optimizer described in Section 4.1. For inference, we recorded short exchanges from speakers not seen during training, serving as prompts for our models, and we created text transcripts that are continuations of the prompts. We then feed the transcripts to the text-to-semantic model, the output of which is fed to SoundStorm, while using the speaker prompts for both stages.

We find that this approach generates high-quality, natural dialogue sequences, generating disfluencies at the occurrence of filler words in the transcript, and allowing for controlled speaker turns through the insertion of the turn marker symbols in the transcript. The total runtime for synthesizing a segment of 30 seconds segment is 2 seconds. We invite the reader to listen to the generated samples on the accompanying webpage in the supplementary material.

## 6 CONCLUSION

In this paper we present SoundStorm, a model that can synthesize high-quality audio from discrete conditioning tokens efficiently. When compared to the acoustic generator of AudioLM, SoundStorm is two orders of magnitude faster and achieves higher temporal consistency when generating long audio samples. By combining a text-to-semantic token model similar to SPEAR-TTS with SoundStorm, we can scale text-to-speech synthesis to longer contexts and generate natural dialogues with multiple speaker turns, controlling both the voices of the speakers and the generated content.

### BROADER IMPACT

SoundStorm is a model for high-quality, efficient generation of neural audio codec-derived representations of audio. In this work, we use it as a replacement for the acoustic generation pipeline of AudioLM and SPEAR-TTS. We acknowledge that the audio samples produced by the model may be influenced by the biases present in the training data, for instance in terms of represented accents and voice characteristics. In our generated samples, we demonstrate that we can reliably control speaker characteristics via prompting. However, a more thorough analysis of any training data and its limitations is an area of future work.

In turn, the ability to mimic a voice can have numerous malicious applications, including bypassing biometric identification and for the purpose of impersonation. Thus, it is crucial to put in place safeguards against potential misuse: to this end, we have verified that the audio generated by SoundStorm remains detectable by a dedicated classifier (98.5% using the same classifier as Borsos et al. (2023)). Hence, as a component of a larger system, we believe that SoundStorm would be unlikely to introduce additional risks to those discussed previously by Borsos et al. (2023) and Kharitonov et al. (2023). At the same time, relaxing the memory and computational requirements of AudioLM would make research in the domain of audio generation more accessible to a wider community.

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

## Task Description

The purpose of this task is to rate the **quality** of speech samples.

- You will listen to short speech samples (8-10 seconds).
- For each speech sample, we define two segments:
  - the first segment, consisting of the first 3 seconds of the sample,
  - the second segment, consisting of the rest of the sample.
- You are asked to rate the quality of the **second segment,** based on:
  - intelligibility of the speech
  - naturalness of the prosody
  - acoustic quality
  - noticeable drifts in the speaker's voice or recording conditions

Figure 5: The instructions for the raters participating in the subjective quality evaluation test.

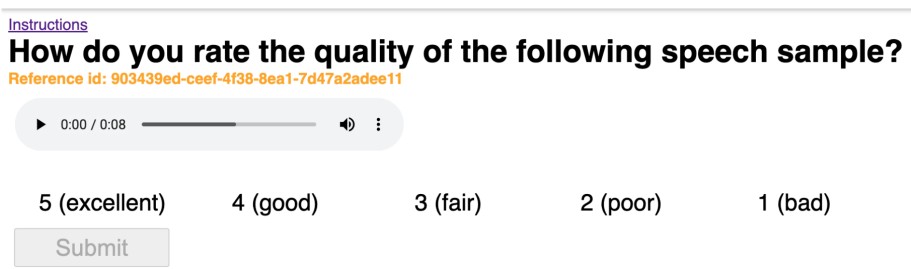

Figure 6: The rater interface.

## A    APPENDIX A: SUBJECTIVE QUALITY EVALUATION TEST

For the subjective quality evaluation test, the raters received the instructions presented in Figure 5. The raters submitted their ratings using the interface shown in Figure 6.

