# OpenReview forum: "SoundStorm: Efficient Parallel Audio Generation"
_ICLR.cc/2024/Conference — Submitted to ICLR 2024_

### Official Review · Reviewer_joNq · 2023-10-20

**Soundness:** 2 fair
**Presentation:** 3 good
**Contribution:** 2 fair
**Rating:** 5
**Confidence:** 4

**Summary:**

The paper proposes a parallel-decoding approach called SoundStorm for generating acoustic tokens given semantic conditional signals or speaker prompts. The core is an extension of the MaskGIT in the scenarios of multiple residual vector-quantizers and in the audio domain. The experiments show that the proposed non-autoregressive method could run faster at the inference stage than previous multi-stage acoustic tokens modeling.

**Strengths:**

1) Using parallel decoding for audio generation is promising and demanding due to the length and RVQs. It will be a growing field shortly.
2) The audio samples in supplementary material are well done.
3) In general, the paper is straightforward to understand.

**Weaknesses:**

1) Technical novelty is quite limited. The masking and decoding algorithms are adapted from MaskGIT. Probably the main difference is that SoundStorm enables the mask on multiple RVQ levels.
2) It is okay if the simple adaptation works the best, but it requires more experiments and comparisons to show that. The current baseline comparisons are very limited. What about the other autoregressive methods mentioned in the related work, like VALL-E and delayed patterns? There is not even comparison to the basic masking approach that masks tokens at arbitrary time steps and RVQ levels randomly. The VampNet could achieve reasonable results with a simple masking approach to music generation.
3) It seems like the speaker prompts highly influence the quality metrics (WER, CER, audio quality). I am not sure why it has to be the case that no tokens of the prompt are masked during training.
4) It is unclear why Conformer is being used instead of the same architecture as AudioLM; no comparison is shown.
5) Reproducibility: most of the components in the proposed approach and even baseline AudioLM are not open-source. While I understand this is not controllable for certain reasons, more implementation details, e.g., optimizers, learning rate, batch size, the cross-entropy loss when convergence, etc., should be included in the paper so that there is a higher chance of reproducing the results and have a fair comparison for future work.

**Questions:**

Besides questions mentioned in weaknesses,
1) The number of RVQ levels affects the number of heads. Usually, more heads take more time to converge. Are there any insights into the relationship between the number of RVQ levels and the convergence in the training?
2) How important are the conditional semantic tokens in the proposed method? Would it still be possible to train with other types of semantic tokens?

---

> ### Author Response · Authors · 2023-11-20
>
> >Weaknesses:
> 1. While some of the building blocks of our model come from previous works, we would like to emphasize that, in contrast to previous works, we show for the first time that it is possible to generate all tokens of a residual vector-quantized neural audio codec non-autoregressively based on conditioning signals such as the semantic tokens with no quality loss.
> 2. We note that none of the cited baselines was proposed for the problem we tackle, i.e., audio generation based on semantic tokens. In terms of comparing other masking strategies for our problem, we were not successful in adopting VampNet's random masking scheme for speech; considering the hybrid scheme of VALL-E, our method is fully non-autoregressive with significant performance gains and no quality loss compared to groundtruth. The delay pattern has been shown to be suboptimal compared to flattened generation [1], and the flattened generation is present as a baseline ("AudioLM" in Table 1 and "AudioLM flattened" in Table 2).
> 3. Speaker prompts do affect the quality metrics because the first 3 seconds of prompted audio is original instead of generated, which naturally results in lower WER.
> 4. We are planning to add a comparison to a Transformer backbone.
> 5. Thank you for pointing this out, we added these details for the revision.
>
> > Questions:
> 1. The number of RVQ will not only impact the training time, but also the generation quality and inference decoding steps – is an interesting avenue we are planning to investigate.
> 2. The conditioning information is very important for our method. Increasing the amount of information makes the synthesis task easier, while, on the other hand, we could not achieve promising results when replacing the semantic tokens with the transcripts.
>
> [1] Copet, Jade, et al. "Simple and Controllable Music Generation." arXiv preprint arXiv:2306.05284 (2023).

---

### Official Review · Reviewer_A5iV · 2023-11-01

**Soundness:** 3 good
**Presentation:** 3 good
**Contribution:** 2 fair
**Rating:** 6
**Confidence:** 3

**Summary:**

The paper introduces SoundStorm, an iterative generative method for semantic-to-acoustic audio tokens generation. It employs the MaskGIT decoding scheme for residual vector quantized tokens. In comparison to the autoregressive (AR) approach of AudioLM, SoundStorm produces audio of comparable quality. Additionally, it offers increased consistency in voice and acoustic conditions, while achieving sampling speeds that are two orders of magnitude faster.

**Strengths:**

* The proposed method not only matches the audio quality of a state-of-the-art autoregressive method but also excels in intelligibility, voice preservation, acoustic consistency, and sampling speed.
* The authors successfully apply the MaskGIT method to the residual vector quantization (RVQ) in the speech domain.

**Weaknesses:**

The performance evaluation focuses solely on SoundStream tokenization and only offers a comparison with AudioLM. While the superiority over stages 2 and 3 of AudioLM is evident, the paper does not elucidate the proposed method's applicability to other neural audio codecs or how it compares with other generative methods. It would be beneficial to explore its viability with tokenization methods other than SoundStream and to contrast it with other generative methods. For instance, comparing it with the hybrid approach of VALL-E, which employs autoregressive modeling at the first level of RVQ and non-autoregressive modeling in subsequent levels, or evaluating against non-autoregressive methods like diffusion [1] or flow-matching [2], could help illuminate potential trade-offs.

[1] Shen, Kai, et al. "Naturalspeech 2: Latent diffusion models are natural and zero-shot speech and singing synthesizers." arXiv preprint arXiv:2304.09116 (2023).

[2] Le, Matthew, et al. "Voicebox: Text-guided multilingual universal speech generation at scale." arXiv preprint arXiv:2306.15687 (2023).

**Questions:**

Although the samples in the supplementary material exhibit generation diversity, is there a way to verify that this diversity stems from the proposed method rather than the semantic token generator? In other words, exploring the diversity of samples generated from the same semantic tokens would also be an interesting aspect of this research.

---

> ### Author Response · Authors · 2023-11-20
>
> > Weaknesses:
>
> In this paper, as you note, we focus on speeding up the acoustic generation stage of AudioLM, by showing matching quality but 100x reduction in the inference time. Our motivation for formulating the claim this way is the immediate consequence that SoundStorm can be used as a plug-in replacement for the models relying on the AudioLM framework without quality loss. For example, SoundStorm can be used as the acoustic generator of SPEAR-TTS [1], which has been shown to significantly improve over VALL-E.
> Regarding the compatibility of SoundStorm with other neural audio codecs, SoundStorm is designed to support not only SoundStream, but any residual vector-quantized autoencoder – we updated the paper to reflect this point.
>
> > Questions:
>
> The samples in the supplementary material relied on synthesis  groundtruth semantic tokens: given the same semantic tokens, SoundStorm will sample different speakers. To quantify this, we will measure the entropy of a speaker classifier given multiple samples from the same semantic tokens.
>
> [1] Kharitonov, Eugene, et al. "Speak, read and prompt: High-fidelity text-to-speech with minimal supervision." arXiv preprint arXiv:2302.03540 (2023).

---

> > ### Comment · Reviewer_A5iV · 2023-11-23
> > **Response to Authors**
> >
> > I appreciate the authors for responding to my comments.  I have no additional questions.

---

### Official Review · Reviewer_9y1z · 2023-11-02

**Soundness:** 3 good
**Presentation:** 2 fair
**Contribution:** 3 good
**Rating:** 5
**Confidence:** 5

**Summary:**

This paper introduces an approach to efficient parallel audio generation. The authors leverage bidirectional attention and confidence-based parallel decoding to sample the tokens of a neural audio codec. Compared to AR approaches, SoundStorm is much more efficient while preserving generation quality.

**Strengths:**

This paper proposes an efficient and high-quality generation diagram for neural audio codec token generation. Compared with the auto-regressive generation, SoundStorm successfully addresses the high-latency issue while enjoying the same quality and even more consistency.

The samples in supplementary material show impressive generation performance, especially the dialogue generation.

**Weaknesses:**

1)	The experiments are not convincing. The authors only compare the proposed SoundStorm with the AudioLM baseline. However, there are many works such as VALL-E [1] and NaturalSpeech 2 [2] successfully addressed the high-quality zero-shot TTS task. More comparison and discussion should be included to demonstrate the generation performance. Furthermore, prosody is also an important aspect of generation quality. Evaluation of prosody should also be conducted.

2)	The human evaluation should be more detailed. In this paper, only MOS is reported. To measure the generation similarity, SMOS results should be reported.

3)	In the human evaluation test (Table 2), the author uses ground-truth semantic tokens, which is not fair for baselines.

4)	Some details are missing to reproduce the results easily.
a)	What are your temperature and top-k values for sampling masked position and sampling tokens in a confidence-based sampling scheme?
b)	Do the “mask” tokens in different RVQ layers share the same embedding?

5)	Related work coverage: Some works such as NaturalSpeech 2 [2] also use an efficient framework (latent diffusion) to model the tokens of neural audio codecs.



[1] Wang, Chengyi, et al. "Neural codec language models are zero-shot text to speech synthesizers." arXiv preprint arXiv:2301.02111 (2023).

[2] Shen, Kai, et al. "Naturalspeech 2: Latent diffusion models are natural and zero-shot speech and singing synthesizers." arXiv preprint arXiv:2304.09116 (2023).

**Questions:**

Firstly, please refer to the Weaknesses part to see most of my questions.

1)	In addition, since SoundStorm is a more efficient and high-quality alternative compared with auto-regressive approaches, I have the following questions:
a)	Can SoundStorm maintain the features of the prompt such as the acoustic environment and the speaker’s emotion?
b)	Since SoundStorm has a sampling mechanism in the acoustic token generation, have you tried the diversity of generated speeches? For example, an AR model can generate different prosody while preserving similar timbre, content, etc.
c)	How does the prompt length affect the generation quality?

2)	Some questions for Table 1:
a)	The WavLM similarity for Codec reconstruction is 0.63 – 0.66, which is low. Does it mean the Codec reconstruction quality is not good?
b)	For Voice preservation and Acoustic consistency results with a speaker prompt setting, do you use a prompt reconstructed by audio codec as an evaluation reference or an original prompt (which means the reference prompt is not constructed by codec)?

---

> ### Author Response · Authors · 2023-11-20
>
> >Weaknesses:
> 1. Our main claim is that we can speed up the AudioLM framework by 100x without loss of quality. Accordingly, all our experimental evaluations are designed for verifying this claim and ablating the design choices. Our purpose with this framing is that our method is directly applicable without loss of quality to all models relying on the AudioLM framework, including SPEAR-TTS [1] which has been shown to significantly improve over VALL-E. We agree regarding the importance of prosody, which is also captured by our MOS study — Appendix A shows that the naturalness of the prosody is one of the criteria for rating.
> 2. Our subjective listening study does capture the generation similarity —  Fig. 6 in Appendix A shows that the noticeable drifts in the speakers’ voices is one of the criteria for rating.
> 3. We are unsure what is meant by “uses ground-truth semantic tokens, which is not fair for baselines”. The gold baseline (“Ground Truth”) in Table 2 is the SoundStream reconstruction of the original audio, whereas the other methods are evaluated for the same task they are trained for, that is, audio synthesis based on the semantic tokens extracted from the original audio.
> 4. Thank you for pointing these out, we added these details to the revised version.
> 5. We added the suggested reference, although we note that NaturalSpeech 2 models the quantized continuous latents, not the discrete tokens.
>
> >Questions:
> 1. a) SoundStorm is conditioned on the semantic tokens. Hence the answer to this question is highly dependent on the type of information the semantic tokens contain. For example, the semantic tokens used in our paper do not fully capture the speaker id, but as shown in the MOS study, SoundStorm manages to maintain the speaker id of the prompt. The speaker's emotion is mostly contained in the semantic tokens.
> b) Given the same semantic tokens, similar to AudioLM’s original acoustic generator, SoundStorm will sample the same semantic content but with different speakers and prosody variations – we provided samples from the same semantic tokens in the supplementary materials.  c) We have not conducted a study on the effect of the prompt length, but it is an interesting direction for future work.
> 2. a) The SoundStream codec used in the paper (6kbps) provides close-to-perceptually lossless reconstruction for speech, however, the similarity score model is not perfectly robust to imperceptible codec artifacts, resulting in lower speaker similarity scores. Nevertheless, we find that the scores correlate strongly with subjective listening test ratings. b) We use the reconstructed prompts.
>
> [1] Kharitonov, Eugene, et al. "Speak, read and prompt: High-fidelity text-to-speech with minimal supervision." arXiv preprint arXiv:2302.03540 (2023).

---

### Official Review · Reviewer_ZBFp · 2023-11-03

**Soundness:** 3 good
**Presentation:** 3 good
**Contribution:** 1 poor
**Rating:** 3
**Confidence:** 3

**Summary:**

This paper proposes to combine the Conformer model architecture (Gulati et al., 2020) with a coarse-to-fine MaskGIT modeling objective (Chang et al., 2022) and apply these methods to model residual quantized representations of audio. This system (collectively called SoundStorm) enables efficient sampling of (long) tokenized audio sequences. Quantitatively, sampling is 2 orders of magnitude faster than baseline autoregressive sampling from the AudioLM (a causally masked autoregressive transformer). A SoundStorm model is trained on the LibriLight dataset for speech synthesis. The quality of this model compares favorably vs. the baseline AudioLM according to automatic eval metrics (Table 1) and human evaluation (Table 2).

**Strengths:**

The experiments convincingly demonstrate that this model outperforms the quality of the baseline AudioLM, with dramatically faster sampling. The decoding steps ablation (Figure 3) is helpful for understanding the tradeoff between sample quality and inference time for this family of models. This tradeoff seems quite favorable, becoming only mildly worse at longer sequence lengths (some dependence on sequence in the quality/iterations tradeoff is to be expected).

**Weaknesses:**

The paper does not articulate a clear contribution. Both the modeling objective (MaskGIT) as well as the architecture (Conformer) are borrowed from previous work; combining these two ideas in itself does not seem like a particularly significant contribution.

There appears to be some novelty in the proposed masking scheme, but the decisions behind this proposal are not thoroughly discussed or empirically ablated. The description of the masking protocol itself is a little hard to follow, in part because the RQV tokenization is never formally defined.

The only empirical comparison to other audio models is AudioLM. I wouldn't consider this a weakness in itself, but given the lack of substantive modeling contributions, I might expect to see a more thorough evaluation. How does SoundStorm compare to, e.g., AudioGen? Is the claim that SoundStorm is simply in a class of its own as far as quality & efficient inference is concerned? If this is indeed the claim, then it could be articulated more clearly with a direct comparison to another NAR speech generation model.

I'm not sure what to make of the results on dialogue synthesis. There is no effort to evaluate these results, merely a pointer to the supplemental material. I'm happy to attest that the supplement results do sound nice, but I'm not sure what I'm supposed to take away from this section or how it fits into the broader claimed contributions of the paper; this reads more like a product advertisement than a rigorous study.

**Questions:**

Do the authors plan to release trained SoundStorm model weights? The clearest contribution of this paper is the model artifact itself; if this model is going to be released then that would help to clarify the contribution of this paper to the community.

---

> ### Author Response · Authors · 2023-11-20
>
> **Contributions**: While some of the building blocks of our model come from previous works, we would like to emphasize that, in contrast those, we show for the first time that it is possible to generate all tokens of a neural audio codec non-autoregressively, using semantic tokens as conditioning signal. The main contribution of the paper is to speed up the generation within the AudioLM framework by 100x without quality loss, which we showcase for speech.
>
> **Baselines and Ablations**: Consequently, given the novel use-case, there is a large number of possible ablations that could be conducted: we prioritized the most crucial ones regarding speech intelligibility, quality and per-level decoding steps and we are planning to add a comparison to Transformer backbone. Regarding comparisons to other speech synthesis baselines, we showed that we can match the quality of AudioLM with significant speedup, hence our method is applicable in all setups relying on AudioLM: for example, our method also applies without quality loss to SPEAR-TTS [1], which has shown to outperform the partially NAR-based VALL-E [2]. We left the non-speech applications (such as MusicLM [3]) for future work.
>
> **Presentation**: Thank you for the observation on the presentation of RVQ, we streamlined it together with the RVQ-motivated design choices. We have also added further details to the revised version related to the training setup for reproducibility.
> The dialogue experiments illustrate the fact that the model also works on domains that are harder to model than single-speaker speech. Due to the lack of well-established metrics on dialogue generation, we left the evaluation of this use-case as future work.
>
> **Model Release**: While we are not planning to release the model weights at this point, but we added all necessary details for reproduction in the revised version.
>
> [1] Kharitonov, Eugene, et al. "Speak, read and prompt: High-fidelity text-to-speech with minimal supervision." arXiv preprint arXiv:2302.03540 (2023).
>
> [2] Wang, Chengyi, et al. "Neural codec language models are zero-shot text to speech synthesizers." arXiv preprint arXiv:2301.02111 (2023).
>
> [3] Agostinelli, Andrea, et al. "MusicLM: Generating music from text." arXiv preprint arXiv:2301.11325 (2023).

---

### Official Review · Reviewer_kvR8 · 2023-11-06

**Soundness:** 4 excellent
**Presentation:** 3 good
**Contribution:** 3 good
**Rating:** 8
**Confidence:** 5

**Summary:**

This paper proposed a parallel audio generation method called SoundStorm which produces high quality coherent speech signals while reducing the generation time two orders of magnitude compared to the autoregressive-based audio generation method AudioLM. This method makes the long audio generation feasible. The proposed method is inspired by an image generation algorithm MaskGIT which has been demonstrated effective for image generation. SoundStorm performs parallel decoding for each RVQ layer to generate the SoundStream codec tokens, proceeding from lower coarse layers to higher finer layers, layer by layer. Conformer layers are applied to model the temporal correlation of the acoustic embeddings. Within each RVQ layer, multiple iterations of inference are performed to estimate the masked portions gradually until all the tokens are estimated. For each iteration, the masks are obtained using the masking schedule (cosine schedule) combined with the confidence scores. Their experiments results demonstrate competitive performance in terms of both subjective and objective quality of the generated speech. Besides generating the single speaker utterance level of speech signal, the paper also conducted an experiment to generate the two-speaker conversational speech with an impressive speech quality and smooth speaker turn.

**Strengths:**

1, While not originally proposing, the paper adopted the MaskGIT method from image domain for efficient parallel audio generation. This method can generate high quality speech signal while requiring much smaller number of inference steps compared to the autoregressive style baseline system AudioLM. The proposed technology allows generating high quality long form audio signals. The proposed method shows great potential to solve such challenging problem.

2, The experimental parts are comprehensive, various evaluations are performed to measure the quality of the generated speech as well as the run time complexity. The results clearly demonstrate the competitiveness of the proposed method.

3, The motivation and background introduction is comprehensive.

**Weaknesses:**

The main weakness of the paper is the writing part which lacks some details someties, or the description is not clear without reading the MaskGIT reference paper. Some examples are listed as below, please revise these parts accordingly.

1, In the 1st paragraph of Sec. 3.3, the definition or the criteria for confidence score should be explained explicitly. As the confidence score is one of the most important key points of the proposed method, it is important to describe it clearly in this section so that the readers do not have to turn to the reference paper.

2, In the 2nd paragraph of Sec. 3.3, when you mention “the conditional independence assumption in finer levels”, you should add that this assumption is made along the time dimension. Otherwise, it could confuse the readers that such conditional independence exists along the RVQ level dimension which conflicts with the fact that inference is performed from lower level to higher level.

3, In sec. 4, the experiments part lack of the training configuration details, such as batch size, learning rate scheduler, optimization method, etc. This happens for both utterance-based generation and conversational speech generation. This leads the proposed work not reproducible.

**Questions:**

1, In Sec. 3, for the ablation study of number of decoding steps, did you also perform such experiment to measure the effect of the number of decoding steps on the WER/CER performance?

2, In Sec. 3.3, iterative parallel decoding, have you tried to replace the unmasked tokens from previous inference stage with the estimation from current inference stage with a higher confidence score? The question here applies to both current RVQ layer and previous RVQ layers.

3, The original MaskGIT paper describes the limitation and failure cases of the MaskGIT method, such as semantic and color shifts, ignore and modify objects on the boundary when applied to outpainting and inpainting, oversmoothing or creates undesired artifacts on complex structure, etc. Are these limitations also appliable to speech generation task? Could you comment the technique limitations of the SoundStorm method?

---

> ### Author Response · Authors · 2023-11-20
>
> > Weaknesses
>
> 1-3. Thank you for pointing these out, we addressed them in the revised version.
>
> >Questions
> 1. Our MOS study captures this aspect partially, where the ratings (which also capture speech intelligibility) do not increase by using more iterations. We will also add the automated metrics.
> 2. We did not try this, but we agree that this is a promising future direction.
> 3. In contrast to MaskGIT, which was proposed with direct text conditioning, we proposed SoundStorm with semantic token conditioning. Arguably, our setup is “easier” in the sense that our semantic tokens already contain a significant amount of information (phonetic content, partial prosody information, some speaker information), which leaves less potential for failures when used as conditioning. This also suggests the limitation of SoundStorm: extending it to weaker conditioning signals such as text could be non-trivial.

---

### Author Response · Authors · 2023-11-20

We thank all reviewers for their time and feedback. We revised the paper around the presentation of RVQ and confidence scores, and added further details about the training setup and hyperparameters.

---

### Meta-Review · Area_Chair_LN1D · 2023-12-04

**Metareview:**

The paper introduces SoundStorm, an iterative generative method that converts semantic tokens into acoustic audio tokens. It employs the MaskGIT decoding scheme for this transformation. When compared to the autoregressive (AR) approach of AudioLM, SoundStorm produces audio of comparable quality. It also enhances consistency in voice and acoustic conditions and achieves faster sampling speeds. However, reviewers have expressed concerns about limited innovation, the need for clarification on contributions, and details of the implementation.

**Justification For Why Not Higher Score:**

N/A

**Justification For Why Not Lower Score:**

N/A

---

### Decision · Program_Chairs · 2024-01-16

Reject